# Are School Substance Use Policy Violation Disciplinary Consequences Associated with Student Engagement in Cannabis?

**DOI:** 10.3390/ijerph17155549

**Published:** 2020-07-31

**Authors:** Megan Magier, Karen A. Patte, Katelyn Battista, Adam G. Cole, Scott T. Leatherdale

**Affiliations:** 1Faculty of Applied Health Sciences, Brock University, 1812 Sir Issac Brock Way, St. Catharines, ON L2S 3A1, Canada; kpatte@brocku.ca; 2Faculty of Applied Health Sciences, University of Waterloo, Waterloo, ON N2L 3G1, Canada; kbattista@uwaterloo.ca (K.B.); sleatherdale@uwaterloo.ca (S.T.L.); 3Faculty of Health Sciences, Ontario Tech University, Oshawa ON L1G 0C5, Canada; adam.cole@uoit.ca

**Keywords:** cannabis, policy, disciplinary consequences, schools, youth

## Abstract

Schools are increasingly concerned about student cannabis use with the recent legalization in Canada; however, little is known about how to effectively intervene when students violate school substance use policies. The purpose of this study is to assess the disciplinary approaches present in secondary schools prior to cannabis legalization and examine associations with youth cannabis use. This study used Year 6 (2017/2018) data from the COMPASS (Cannabis use, Obesity, Mental Health, Physical Activity, Alcohol use, Smoking, Sedentary behavior) study including 66,434 students in grades 9 through 12 and the 122 secondary schools they attend in British Columbia, Alberta, Ontario, and Quebec. Student questionnaires assessed youth cannabis use and school administrator surveys assessed potential use of 14 cannabis use policy violation disciplinary consequences through a (“check all that apply”) question. Regression models tested the association between school disciplinary approaches and student cannabis use with student- (grade, sex, ethnicity, tobacco use, binge drinking) and school-level covariates (province, school area household median income). For first-offence violations of school cannabis policies, the vast majority of schools selected confiscating the product (93%), informing parents (93%), alerting police (80%), and suspending students from school (85%), among their disciplinary response options. Few schools indicated requiring students to help around the school (5%), issuing a fine (7%), or assigning additional class work (8%) as potential consequences. The mean number of total first-offence consequences selected by schools was 7.23 (SD = 2.14). Overall, 92% of schools reported always using a progressive disciplinary approach in which sanctions get stronger with subsequent violations. Students were less likely to report current cannabis use if they attended schools that indicated assigning additional class work (OR 0.57, 95% CI (0.38, 0.84)) or alerting the police (OR 0.81, 95% CI (0.67, 0.98)) among their potential first-offence consequences, or reported always using the progressive discipline approach (OR 0.77, 95% CI (0.62, 0.96)) for subsequent cannabis policy violations. In conclusion, results reveal the school disciplinary context in regard to cannabis policy violations in the year immediately preceding legalization. Various consequences for cannabis policy violations were being used by schools, yet negligible association resulted between the type of first-offence consequences included in a school’s range of disciplinary approaches and student cannabis use.

## 1. Introduction

On 17 October 2018, Canada implemented Bill C-45 to legalize and regulate recreational cannabis consumption among adults [1]. One of the main intentions of Bill C-45 is to reduce youth access and deter early-onset of use [2]. According to the Canadian Student Tobacco, Alcohol and Drugs Survey in 2016/2017, 17% of youth in grades 7–12 reported past 12-month cannabis use, with grade 12 students having the highest reported usage at 34.5% [1]. Canadian youth have been identified as having some of the highest rates of past-year use when compared to other countries [2]. With the recent legalization of cannabis, substance use prevention targeting youth has become a strategic priority. 

Schools are key contexts for equitable prevention strategies as the location where almost all youth, regardless of socioeconomic status (SES), spend approximately 25 h a week during the school year. School-based approaches have the potential to prevent cannabis use among students [3], which may, in turn, protect their educational attainment, and cognitive, mental, and physical health [4]. Cannabis use in youth is associated with an increased likelihood of disengagement from school, drop out, and lower achievement levels [4]. Conversely, studies have shown that achievement in academia and engagement in school provide protective measures against substance use [5]. Furthermore, school-wide social norms and school climate are associated with student substance use [6]. Some evidence suggests that students who attend schools with a positive climate feel supported and encouraged, are more engaged in academics and school activities, and are less likely to engage in risk behaviors [7]. That is, a supportive school environment may act as a protective mechanism for youth against a multitude of problem behaviors, including substance use [8]. 

School climate may be influenced by school policies and disciplinary approaches. Although schools are not expected to prevent all substance use among students, school policies have the potential to restrict or prevent student substance use during school h and on school property; however, limited research has examined how to effectively respond when students violate these policies. Most schools have consequence measures for the use of drugs on school property or during school h, although differences in school-to-school disciplinary approaches exist. There has been a general movement in US and Canadian schools away from more traditional “authoritarian” or punitive disciplinary approaches to more supportive “authoritative” strategies, and similarly from “zero-tolerance” to progressive disciplinary approaches, where sanctions get stronger with each violation [9]. Stemming from Baumrind’s (1968) work on parenting types [10], authoritarian approaches are described as demanding and with no expectation of explanations for actions, whereas authoritative discipline in schools uses structure and support to respect student autonomy [11]. 

Punitive approaches have traditionally been used to scare students into compliance, but some research suggests this can further alienate students that need help, potentially increasing their likelihood of substance use, drop out, and delinquent behavior [12,13]. Furthermore, certain disciplinary consequences, such as out-of-school suspension, have been associated with unintended negative consequences and appear largely ineffective as deterrents [13,14,15]. For instance, students attending schools that used out-of-school suspension for substance use violations were 1.6 times more likely to use cannabis compared to those who attended schools not using out-of-school suspensions. Historically, “scare them straight” education programs, such as “Just Say No” campaigns and the “DARE” program have produced disappointing results [16]. These universal programs are often more economical and require less field expertise, however, factors such as youth experimentation and peer influences often cause them to disengage from these types of preventative measures [16]. Student Assistance Programs (SAP) demonstrate more success and have been linked to positive school climates [16]. SAPs coordinate preventative services with school-based services, ranging from educational to remediation and counselling programs. These less punitive approaches, such as counselling, have been shown to predict reductions in the likelihood of later cannabis use by an average of 50% [3].

In 2009, the Ontario Ministry of Education mandated all schools to have a Progressive Discipline Policy, which is designed to correct inappropriate behaviors in students and promote more positive actions and decision making, by shifting from a solely punitive approach to one that provides learning opportunities [17]. Based on this policy, schools should consider a range of supports (e.g., counselling) and consequences (e.g., an assignment, detention) to determine the most appropriate response for each situation that will help students learn from their choices and reflect on the impact [18]. City- and board-wide versions of this policy have occurred in other locations throughout Canada (e.g., the Calgary Board of Education in Alberta) [9]. Apart from these developments, little is known about what disciplinary approaches schools have implemented and how they relate to student risk behaviors, such as substance use. If effective, changing school policies and disciplinary approaches offers a low cost and pragmatic way to help decrease substance use [14]. It is important to investigate what disciplinary approaches are currently in place within the Canadian secondary school system and associations with student cannabis use in order to support evidence-based policy refinement and prevention strategies. The aim of this study is to assess a range of potential disciplinary consequences being used by secondary schools in response to student violations of school cannabis policies in the year immediately preceding cannabis legalization in Canada (2017/2018) and to examine associations with youth cannabis use. 

## 2. Methods

### 2.1. Design and Sample 

COMPASS (Cannabis use, Obesity, Mental Health, Physical Activity, Alcohol use, Smoking, Sedentary behavior) is an ongoing (2012–2021) longitudinal study designed to collect hierarchical data once annually from students in grades 9 through 12 and the secondary schools they attend [19]. A full description of the COMPASS study is available in print [19] and online (www.compass.uwaterloo.ca). All procedures received ethics approval from the University of Waterloo and Brock University Human Research Ethics Committees and all participating school boards. The present study used cross-sectional student- and school-level data from Year 6 (Y6: 2017/2018) of the study, which included 66,434 students at 122 secondary schools in four Canadian provinces: British Columbia (BC) (*n* = 16), Alberta (*n* = 8), Ontario (*n* = 61), and Quebec (*n* = 37). Notably, Y6 marked the year before cannabis legalization in Canada, providing data about baseline school policies. Schools were purposely selected based on whether they permitted active-information passive-consent protocols, which is critical for collecting robust data on youth substance use [20]. All grade 9 through 12 students attending participating schools were eligible to participate and could decline at any time. Further details of recruitment methods are described elsewhere [21]. The overall student response rate in Y6 was 81.85% of eligible students. Student non-participation primarily resulted from absences or scheduled study-periods during data collection. A complete case sample was used, leaving 60,384 students for the analysis after removing students missing cannabis use data (*n* = 6050). 

### 2.2. Tools 

Student-level data were collected using the COMPASS student questionnaire (Cq), a paper-based survey designed to collect student-reported data from full school samples during one classroom period [19]. School-level data were collected using the COMPASS School Program and Policies (SPP) tool, an online survey designed to assess the presence or absence of policies, practices, and resources relevant to student health behaviors in the school environment [19]. The SPP was completed by the school administrator(s) most knowledgeable about the school program and policy environment. School contacts are encouraged to consult with other staff members and have a small group complete the SPP. Also, COMPASS school knowledge brokers follow up with staff to clarify any unclear or missing data.

## 3. Measures

### 3.1. School Cannabis Policy Violation Disciplinary Approaches

Items on the SPP were used to assess the potential disciplinary approaches used by schools in response to student violations of school substance use policies. Approaches included in a school’s range of disciplinary consequences for *first-offence policy violations* were determined by the question: “What are the consequences for a first offence for students who are caught violating your school’s written policies or practices on marijuana? (Check all that apply)”. Response items included: (a) “Issue warning”; (b) “Inform parents or guardians”; (c) “Refer to a school administrator”; (d) “Refer to a school counsellor”; (e) “Encourage, but not require, to participate in an assistance, education, or cessation program”; (f) “Require to participate in an assistance, education or cessation program”; (g) “Confiscate substance”; (h) “Assign additional class work”; (i) “Assign to help around school”; (j) “Fine”; (k) “Place in Detention”; (l) “Give in-school suspension”; (m) “Suspend from school”; and (n) “Alert police”. Response options vary from authoritarian approaches (i.e., suspension, detention, fine, confiscation, issuing warnings, alert police) [9] to more authoritative approaches (i.e., encourage an assistance, education, or cessation program, refer to a counsellor, helping around the school, additional class work) [9,10]. 

Whether schools use a *progressive disciplinary approach* for subsequent cannabis policy violations was assessed by asking: “Do sanctions get stronger with subsequent violations of marijuana use (i.e., progressive discipline approach)?” Responses were dichotomized as “yes” if schools reported “always” using a progressive discipline approach and “no” if the school reported “sometimes” or “never” using this approach. Since only one school indicated “never” using a progressive disciplinary approach, responses were collapsed into “no” if the school reported “sometimes” or “never”.

### 3.2. Student Cannabis Use 

*Cannabis use* was assessed by asking students: “In the last 12 months, how often did you use marijuana or cannabis?” with the items; (a) “I have never used marijuana”; (b) “I have used marijuana but not in the last 12 months”; (c) “Less than once a month”; (d) “Once a month”; (e) “2 or 3 times a month”; (f) “Once a week”; (g) “2 or 3 times a week”; (h) “4 to 6 times a week”; (i) “Every day”. Current cannabis use was defined as using cannabis at least once a month in the last 12 months; all others (including never users) were considered non-current cannabis users. Cannabis measures are consistent with the national surveillance measures for cannabis use [20].

### 3.3. Covariates

Student-level covariates included: sex (male, female); grade (9, 10, 11, 12, other (Secondary I-II in Quebec)); ethnicity (white, non-white or mixed ethnicity); binge drinking status (non-current binge drinker, current binge drinker (defined as drinking 5 or more alcoholic drinks at least once a month in the last 12 months)); smoking status (non-current smoker, current smoker (defined as smoking tobacco in the last 30 days)). School-level covariates included: province (British Columbia, Alberta, Ontario and Quebec) and school area median household income generated using data from the 2016 Census on census divisions that corresponded with school postal codes [22]. 

## 4. Analyses

Statistical analysis was performed using SAS 9.4. Frequency statistics were calculated on the prevalence of the various disciplinary consequences within schools. Chi-square tests of significance were used to analyze associations between student cannabis use and each of the potential first-offence school substance use policy violation disciplinary consequences on the school administer survey and school-reported use of a progressive discipline approach. Next, school-level descriptives and chi-square tests were computed to compare school first-offence disciplinary response selections by whether schools indicated using a progressive disciplinary approach. To investigate patterns in how the schools completed the disciplinary response measures, and given that alerting the policy is a more severe disciplinary response, and we then compared responses based on their selection of “alert police” among their choice of first-offence approaches. Fischer’s exact test (Freemon–Halton extension for R > 2) was used to account for small cell sizes at the school level.

Multi-level logistic regression models tested the association between the potential disciplinary approaches reported by schools and student cannabis use, with student- and school-level covariates. Models were run using PROC GENMOD, with independent working correlation clustered by school. Additionally, the intraclass correlation coefficient (ICC) was calculated using generalized linear random intercept models to determine the proportion of the variation in student cannabis use explained at the school-level. 

## 5. Results

Table 1 provides sample descriptive statistics and chi-square tests by student-reported current cannabis use and non-current use. More males reported current cannabis use than females, and rates of cannabis use increased with grade. Co-use of cannabis and binge drinking was reported by 41.1% of youth, and 59.7% reported both tobacco and cannabis use. Students from participating Ontario and Alberta schools had the highest reported cannabis use at 14.5% and 14.2%, respectively, compared to students in participating Quebec (7.9%) and British Columbia (8.9%) schools.

### 5.1. School Substance Use Policy Violation Disciplinary Approaches 

Data from the SPP was used to examine the array of disciplinary approaches used by secondary schools in response to student violation of school cannabis policies in the year immediately preceding legalization (see Figure 1). The mean number of total first-offence consequences selected by schools was 7.23 (SD = 2.14), when asked to select all that apply from a list of 14 options. The approach selected by the most schools was to refer the student to a school administrator (96%), followed by confiscation of the substance (93%), and informing the student’s parents (93%). About one-third of schools selected requiring an assistance, education, or cessation program (34%), and 59.8% selected encouraging but not requiring a program, among the potential responses. Over half indicated referring students to a school counsellor (57%). Few schools indicated requiring students to help around the school (5%) or complete additional class work (8%) as potential first-offence disciplinary consequences. 

Always using a progressive disciplinary approach, in which sanctions become stronger with subsequent violations, was reported by 92% of schools. Participating schools in British Columbia and Ontario had the highest reported use of this approach (94% and 97% of schools responded “always”, respectively), compared to schools in Quebec (92%) and Alberta (75%), as indicated by Table 2.

To further understand the patterns in which schools completed the disciplinary response measures, we explored the possibility that schools selecting “alert police” as a potential first-offence consequence tended to choose more options overall, and if in addition to alerting police, these schools also selected more punitive or authoritarian options from the list. Table 3 provides a comparison of schools based on whether “alert police” was selected among the provided response options as a potential consequence for student first-offence violations of school cannabis policies. The mean total of first-offence options selected by schools was higher among those that included “alert police” among their options (7.7, standard deviation = 1.9) in comparison to an average of 5.1 (standard deviation = 2.0) options selected by schools that did not report “alert police” among their choice of disciplinary approaches used (*p* < 0.0001). All 8 schools that selected “issue a fine” also selected “alert police” among their first-offence disciplinary responses. Schools that selected “alert police” also selected out-of-school suspensions (92.9% vs. 54.2%) and requiring (40.8% vs. 8.3%) and encouraging participation in an assistance, education, or cessation program (65.3% vs. 37.5%) more often than schools that did not indicate “alert police”; although differences should be interpreted with caution given small cell counts. 

Next, we explored whether certain first-offence approaches were more likely to be selected by schools that indicated always using a progressive discipline approach to school substance use policy violations, in which sanctions get stronger with subsequent offences. Table 4 provides a comparison of schools based on whether they indicated “always” or “sometimes/never” using a progressive discipline approach. Schools that identified always using this approach selected an average of 7.4 (standard deviation = 2.1) first-offence disciplinary options, compared to 6.3 (standard deviation = 1.7) among the 8 schools that reported sometimes or never using the progressive approach, although the difference was not statistically significant (*p* = 0.13). Comparisons are limited due to small cell counts, with only 8 schools indicating never/sometimes using the progressive disciplinary approach. 

### 5.2. Associations between School Disciplinary Approach Options and Student Cannabis Use

Frequency and chi-square tests of student cannabis use by each potential school disciplinary approach option are reported in Table 5. Student current cannabis use was higher at schools that selected using the following options among their choice of potential first-offence disciplinary approaches: Assigning additional class work, requiring participation in an assistance, education, or cessation program, issuing a warning, and in-school suspensions. Student current cannabis use was lower at schools that selected “alert police” among their array of first-offence discipline options. A random intercept model showed significant between-school variability in the odds of student cannabis use σ^2^_μ0_ = 0.308 (0.046), *p* < 0.0001. The resulting ICC indicated 8.56% of the variance in student-reported cannabis use status is due to differences at the school-level.

Results of the three generalized linear random intercept models are presented in Table 6. In Model 1, school selection of “always” using the progressive discipline approach was tested as the independent variable (vs. “sometimes”/”never”), with current student cannabis use as the dependent variable, adjusting for other student substance use (tobacco, binge drinking) and various sociodemographic variables (province, grade, sex, ethnicity, and school-area median income). Results of Model 1 (without first-offence approaches included) were consistent with Model 3 (with first-offence approaches included), in which school reports of always using progressive discipline were significantly associated with lower student cannabis use. 

In Model 2, school selection of each substance use first-offence policy violation disciplinary approach option were tested as the independent variables, with current student cannabis use as the dependent variable, and adjusting for other covariates. Students attending schools that indicated referring students to a school administrator had an increased likelihood of current cannabis use (OR 1.31, 95% CI (1.04, 1.66)). In contrast, students attending schools that indicated assigning additional class work (OR 0.57, 95% CI (0.38, 0.85)) or alerting police (OR 0.78, 95% CI (0.64, 0.96)) among potential consequences were less likely to report current cannabis use compared to students attending schools that did not select these approaches.

Model 3 results were similar to Model 2, in that students attending schools that indicated assigning additional class work (OR 0.57, 95% CI (0.38, 0.84)) or alerting police (OR 0.81, 95% CI (0.67, 0.98)) among potential consequences were less likely to report current use. However, the association with school selection of referring students to a school administrator as a disciplinary option was no longer significant (OR 1.27, 95% CI (1.00, 1.63)) when school identification of using a progressive discipline approach was not included in the model. Also, when controlling for progressive discipline (Model 3), students attending schools that selected encouraging but not requiring participation in an assistance, education, or cessation program as a disciplinary response option were more likely to report current cannabis (OR 1.16, 95% CI (1.01,1.33)), albeit the effect size was small. No difference in cannabis use was found for students attending schools that selected *requiring* participation in an assistance, education, or cessation program (OR 1.00, 95% CI (0.84,1.19)) or any of the other consequence options provided as first-offence disciplinary responses. As mentioned, students at schools that reported they always used a progressive discipline approach were less likely to report current use of cannabis (OR 0.77, 95% CI (0.62,0.96)) than students at schools that reported sometimes or never using the approach.

## 6. Discussion

This exploratory study assessed the disciplinary approach context in Canadian secondary schools with regards to school cannabis policy violations. Associations between the types of disciplinary response options schools identify using and student cannabis use were also examined. Data were collected from secondary schools in Ontario, Alberta, British Columbia, and Quebec that participated in the COMPASS Study in the year immediately preceding legalization (the 2017/2018 school year). Participating schools identified multiple and varied options for potential consequences for first-offence policy violations, from informing parents to alerting police; however, based on the regression models, few types of disciplinary response options were associated with student cannabis use. Students attending schools that reported always using a progressive disciplinary approach—in which sanctions get stronger with subsequent cannabis policy violations—were less likely to report current cannabis use. 

Students attending schools that reported assigning additional class work as a first-offence policy violation consequence among the approaches used were less likely to report current cannabis use than their peers attending schools that did not report this approach, adjusting for selection of other disciplinary approaches and student characteristics. Assigning additional class work was one of the first-offence responses selected the least frequently among the provided options, with only 8% (10/122) of participating secondary schools indicating using this approach. School selection of “alert police” was more prevalent, with 80% of participating schools reporting this approach among the disciplinary options used in response to student first violations of school policies, and students attending these schools were less likely to report current cannabis use. 

Attending schools that indicated encouraging but not requiring participation in an assistance, education, or cessation program as a response among their options of disciplinary consequences was associated with higher student cannabis use, but only when controlling for reported use of the progressive discipline approach. This strategy may be ineffective in preventing cannabis use in youth, as encouragement may not be enough to persuade and support students in attending programs. However, no association with student cannabis use resulted when not adjusted for school identified always use of a progressive discipline approach, or for schools that reported *requiring* participation in an assistance, education, or cessation program among their first-offence response options. Schools may encounter barriers in finding or knowing what programs to refer students to. Harm reduction initiatives may be considered more acceptable to youth than interventions aimed at abstinence. Students may not feel they need to change this behavior as many youth perceive cannabis use as less harmful than other substances [23]. Reasons behind this belief include: The availability and acceptability of cannabis, and the perceptions of positive effects on managing pain and stress [23]. Interpreting these results further is limited by the current measures. While encouraging or requiring participation in a program may be among the approaches indicated as available for use at these schools, it may not be utilized frequently, potentially due to a lack of program funding or skilled staff. Further research is needed to explore how often and when such approaches are applied by schools. In addition, given the cross-sectional design, it cannot be determined whether inclusion of certain disciplinary approaches among the array of options used by schools contribute to lower or higher student cannabis use, or vice-versa. For instance, higher student cannabis use may lead schools to include encouragement of substance use assistance, education, or cessation programs as responses. 

It should be noted that this study took place in the year immediately preceding legalization, and as such, provides a baseline assessment, with many programs and educational resources continuing to be developed targeting youth cannabis use post legalization. As per the movement away from zero-tolerance and authoritarian tactics, schools may include fewer punitive responses and increasingly draw on more supportive options and a progressive approach in future years. Future studies should explore whether the types of disciplinary approaches used by schools change over time. Furthermore, consistent with previous research [24,25,26], current smoking and binge drinking were associated with increased likelihood of also reporting current cannabis use among students. Future research could investigate what disciplinary approaches are effective in targeting all three substances together.

Overall, 93% of participating schools reported always using a progressive discipline approach for cannabis policy violations, where sanctions get stronger for each offence. In Ontario, the progressive discipline policy became provincially mandated for schools in 2009 [27,28]. Results support high compliance with 97% of participating Ontario schools reporting using this approach for cannabis use. Exploratory results support continued use of this approach. Based on the association with student cannabis use, other schools may want to consider implementing progressive discipline policies. Previous research suggests zero tolerance policies are ineffective and unfair to students, as they provide a blanket approach to discipline [7,11]. Furthermore, they fail to alter the student’s behavior and can exacerbate the negative behavior [15]. There was no statistically significant difference in the number of first-offence response options selected by schools indicating always using a progressive discipline approach, in comparison to those schools that reported sometimes/never. Also, no significant differences resulted in the types of first-offence disciplinary actions selected by school identification of using a progressive discipline approach, yet comparisons are limited with the large majority of schools identifying always’ using this practice. Additional research is needed to identify the specific consequences used by schools that follow a progressive discipline approach to further elucidate the most effective comprehensive strategies. 

Exploratory results of the current study suggest students may be less likely to use cannabis at schools that include more authoritative consequences in their disciplinary response, along with the progressive discipline approach. While cross-sectional, these results compliment previous research indicating that the use of less punitive and more supportive approaches are associated with less student cannabis use [3,9,14]. Additionally, consistent with previous research [14], inclusion of out-of-school suspensions and more punitive approaches among potential consequences were not associated with student cannabis use, with the exception of alerting the police. Contrary to expectations, students attending schools that selected alerting the police as one of their potential first-offence consequences were less likely to report current cannabis use than their peers attending schools that did not report this approach. In contrast, previous studies have suggested that more punitive approaches that aim to scare youth into compliance may later increase their substance use [3,9,14]. Students may see police officers as having more power and authority in the judicial system than teachers [26] and cannabis remains illegal for youth to use. It should be noted that alerting the police is not a legal requirement of schools by the Federal Government in Canada. For matters pertaining to cannabis violations, provinces, such as Ontario, suggest principals consider mitigating factors when deciding whether to alert the police in these discretionary situations [29]. 

Alerting the police is not a requirement of schools when they catch students violating school substance use policies in Canada, even in the case of cannabis. It is often up to the discretion of the principal to decide whether to involve the police or not. Given that alerting the police is a more severe disciplinary response, we explored the possibility that schools that selected this option may be different from those that did not. Schools that reported “alert police” as a first-offence disciplinary response to school cannabis policy tended to select more options overall in response to the question. These schools also indicated out-of-school suspension among their first-offence responses more often, suggesting some schools may have an overall more punitive or authoritarian context. However, interestingly, therapeutic approaches of requiring or encouraging students to participate in a program were also reported more frequently by schools that selected “alert police”. As mentioned, it is not known how often or under what circumstances each approach indicated is implemented by schools. It is plausible that police are infrequently involved, and only remain among the set of options selected by some schools from times when more punitive tactics were common. Further research on this approach is necessary, including more comprehensive examination of potential adverse effects over time. 

### Limitations and Strengths

A key strength of this study is the large sample size, with data at both the student- and school-levels in four Canadian provinces, although a number of limitations warrant consideration. The use of student and school administrator-reported measures creates the possibility of recall and social desirability biases. There is a concern of under-reporting cannabis use in students, notably for females, for whom use is more stigmatized [29]. However, the COMPASS study uses active-information passive-consent protocols and does not require student names, helping to reduce selection bias and to preserve perceptions of confidentiality and anonymity. While the school surveys are to be completed by the school contact(s) most knowledgeable about the school health program and policy environment, it is plausible that the respondents are unaware of how policies are being implemented. Given that the question assessing first offence approaches used a “select all that apply” design, the responses may not accurately reflect the usual consequences used by school administrators. That is, the data do not allow analysis of *how frequently* each first-offence disciplinary approach selected by the schools are utilized or how consequences for each case are decided on among the potential options. It is likely that school administrators take many factors into account when deciding on appropriate responses to cannabis policy violations, and future qualitative research should explore how these decisions are made. Additionally, the implementation of substance use prevention or cessation programs in schools was not controlled for in the current exploratory analysis. Prevention programs have the potential to decrease student-reported cannabis use within a school and should be considered in the context of disciplinary approaches in future research. Furthermore, interpretations are limited by the use of cross-sectional data, which does not allow evaluation of the effectiveness of school disciplinary approach environments on deterring student use. As discussed, schools may be more likely to use certain disciplinary approaches in response to higher prevalence of student cannabis use. Lastly, while the large sample supports generalizability, the COMPASS study was not designed to be provincially or nationally representative.

## 7. Conclusions

Examining associations with school disciplinary approach environments is critical to avoid unintended negative consequences, as well provide schools with the ability to make evidence-based policy and practice decisions for student cannabis and other substance use. This research is timely with the recent legalization of cannabis in Canada. Overall, this study provides information on the array of disciplinary approach options used in secondary schools in Canada and associations with student cannabis use in the year preceding legalization. Schools reported using a range of first-offence consequences, from punitive or authoritarian (e.g., suspend from school, detention), restitutive or moderate (e.g., assign help around the school or additional classwork), to more supportive or remedial (e.g., referrals to counselling) approaches. The majority of participating secondary schools in Ontario reported using a progressive discipline approach, in compliance with the mandated policy, as did most participating schools in Alberta, British Columbia, and Quebec. Exploratory results support the use of a progressive disciplinary approach; students attending schools that reported using progressive discipline with increasingly stronger sanctions for subsequent offences were less likely to report cannabis use. Given the negligible differences in student cannabis use by school disciplinary approach options, other factors within the school environment (i.e., social norms and student perceptions) may have a larger influence on cannabis use for students [6,8,29]. Future research examining student perceptions of different school disciplinary contexts and the effectiveness of various approaches over time is warranted. In addition, research involving qualitative interviews with schools to examine how policies are developed, reviewed, and implemented, including how schools select first-offence approaches and why, is an important next step. 

## Figures and Tables

**Figure 1 ijerph-17-05549-f001:**
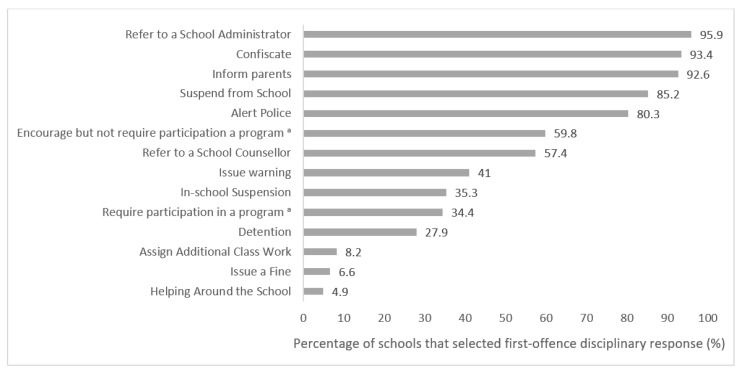
Frequency of various first-offence disciplinary approaches selected by schools (*n* = 122) as being used in response to student violations of school cannabis policies in Year 6 (2017/2018) of the COMPASS study. ^a^ Assistance, education, or cessation program. Note: The question used a “check all that apply” design.

**Table 1 ijerph-17-05549-t001:** Descriptive statistics by student-reported cannabis use in Year 6 (2017/2018) of the COMPASS (Cannabis use, Obesity, Mental Health, Physical Activity, Alcohol use, Smoking, Sedentary behavior) study.

		Non-Current Use (*n* = 53,414)		Current Use (*n* = 6970)		Chi-Square *p*-Value
		*n*	%	*n*	%	
**Province**	Alberta	2605	85.8%	431	14.2%	<0.0001
	British Columbia	10,669	91.1%	1047	8.9%	
	Ontario	24,585	85.5%	4156	14.5%	
	Quebec	15,555	92.1%	1336	7.9%	
**Grade**	9	13,670	93.1%	1017	6.9%	<0.0001
	10	13,071	88.3%	1732	11.7%	
	11	11,792	84.6%	2148	15.4%	
	12	7491	81.1%	1741	18.9%	
	Other ^a^	7390	95.7%	332	4.3%	
**Sex**	Female	27,398	90.4%	2900	9.6%	<0.0001
	Male	26,016	86.5%	4070	13.5%	
**Ethnicity**	White	39,087	88.4%	5111	11.6%	0.7888
	Non-white or mixed	14,327	88.5%	1859	11.9%	
**Binge Drinking**	Non-current	47,376	94.5%	2759	5.5%	<0.0001
	Current	6038	58.9%	4211	41.1%	
**Smoking**	Non-current	51,142	93.4%	3610	6.6%	<0.0001
	Current	2272	40.3%	3360	59.7%	
		Mean	SD	Mean	SD	*t*-test *p*-value
**School Area Median Household Income**		68,952	17,432	70,041	17,033	<0.0001

^a^ Secondary I-II in Quebec schools.

**Table 2 ijerph-17-05549-t002:** Frequency of various disciplinary approach options selected by secondary schools (*n* = 122) as being used in response to student violations of school cannabis policies in Year 6 (2017/2018) of the COMPASS study.

		AB (*n* = 8) % (*n*)	BC (*n* = 16) % (*n*)	ON (*n* = 61) % (*n*)	QC (*n* = 37) % (*n*)
First-Offence Disciplinary Approaches					
Inform parents		100% (8)	94% (15)	92% (56)	92% (34)
Issue fine		0% (0)	0% (0)	13% (8)	0% (0)
Assign additional class work		0% (0)	6% (1)	7% (4)	14% (5)
Require participation in an assistance, education, or cessation program		13% (1)	31% (5)	31% (19)	46% (17)
Assign to help around school		0% (0)	6% (1)	7% (4)	3% (1)
Issue warning		0% (0)	38% (6)	39% (24)	54% (20)
In-school suspension		63% (5)	63% (10)	21% (13)	41% (15)
Suspend from school		75% (6)	75% (12)	89% (54)	86% (32)
Refer to school administrator		100% (8)	100% (16)	90% (58)	95% (35)
Refer to counsellor		50% (4)	88% (14)	69% (42)	27% (10)
Encourage participation in an assistance, education, or cessation program		50% (4)	63% (10)	61% (37)	59% (22)
Confiscate substance		100% (8)	100% (16)	90% (55)	95% (35)
Place in detention		63% (5)	0% (0)	25% (15)	38% (14)
Alert police		75% (6)	63% (10)	80% (49)	89% (33)
Progressive Discipline Approach ^a,b^	(Always)	75% (6)	94% (15)	97% (57)	92% (33)

^a^ 2 Ontario schools did not respond to this question; ^b^ 1 Quebec school did not respond to this question.

**Table 3 ijerph-17-05549-t003:** Comparing schools in Year 6 (2017/2018) of the COMPASS study based on whether “alert police” was selected among options in a “check all that apply” question regarding disciplinary approach responses for first-offence violations of school cannabis policies.

		“Alert Police”				*t*-Test *p*-Value
		Selected (*n* = 78)		Not Selected (*n* = 24)		
Total First-Offence Disciplinary Response Options (Mean [SD])		7.7 (1.9)		5.1 (2.0)		<0.0001
Range of Total First-Offence Disciplinary Response Options Selected		2–14		0–9		-
First-Offence Disciplinary Response Selected		n	%	n	%	Chi-sq *p*-Value *
Inform parents	Selected	92	93.9%	21	87.5%	0.1747
	Not selected	6	6.1%	3	12.5%	
Issue fine	Selected	8	8.2%	0	0.0%	0.1634
	Not selected	90	91.8%	24	100.0%	
Assign additional class work	Selected	8	8.2%	2	8.3%	0.3151
	Not selected	90	91.8%	22	91.7%	
Require participation in an assistance, education, or cessation program	Selected	40	40.8%	2	8.3%	0.0014 *
	Not selected	58	59.2%	22	91.7%	
Assign to help around school	Selected	4	4.1%	2	8.3%	0.2467
	Not selected	94	95.9%	22	91.7%	
Issue warning	Selected	42	42.9%	8	33.3%	0.1309
	Not selected	56	57.1%	16	66.7%	
In-school suspension	Selected	37	37.8%	6	25.0%	0.0994
	Not selected	61	62.2%	18	75.0%	
Suspend from school	Selected	91	92.9%	13	54.2%	<0.0001 *
	Not selected	7	7.1%	11	45.8%	
Refer to school administrator	Selected	95	96.9%	22	91.7%	0.2025
	Not selected	3	3.1%	2	8.3%	
Refer to counsellor	Selected	58	59.2%	12	50.0%	0.1300
	Not selected	40	40.8%	12	50.0%	
Encourage participation in an assistance, education, or cessation program	Selected	64	65.3%	9	37.5%	0.0091 *
	Not selected	34	34.7%	15	62.5%	
Confiscate substance	Selected	94	95.9%	20	83.3%	0.0399 *
	Not selected	4	4.1%	4	16.7%	
Place in detention	Selected	28	28.6%	6	25.0%	0.1927
	Not selected	70	71.4%	18	75.0%	
Progressive discipline approach (sanctions get stronger with each subsequent offence)						
	Always	92	93.9%	19	79.2%	0.1266
	Sometimes/Never	5	5.1%	3	12.5%	
	Missing	1	1.0%	2	8.3%	

***** Interpret results with caution given small cell counts. Fisher’s exact test (Freemon–Halton extension for R > 2) used to account for small cell sizes.

**Table 4 ijerph-17-05549-t004:** Comparing secondary schools in Year 6 (2017/2018) of the COMPASS study based on use of a progressive discipline approach for student violations of school substance use policies.

		Progressive Discipline Approach				*t*-Test *p*-Value
		Sometimes/ Never		Always		
Total First-Offence Disciplinary Response Options (Mean [SD])		6.3 (1.7)		7.4 (2.1)		0.130
Range of Total First-Offence Disciplinary Response Options Selected		3–8		2–14		-
First-Offence Disciplinary Response Selected		n	%	n	%	Chi-sq *p*-Value *
Inform parents	Selected	6	75.0%	105	94.6%	0.0808
	Not selected	2	25.0%	6	5.4%	
Issue fine	Selected	0	0.0%	8	7.2%	0.5630
	Not selected	8	100.0%	103	92.8%	
Assign additional class work	Selected	0	0.0%	10	9.0%	0.3751
	Not selected	8	100.0%	101	91.0%	
Require participation in an assistance, education, or cessation program	Selected	2	25.0%	39	35.1%	0.2686
	Not selected	6	75.0%	72	64.9%	
Assign to help around school	Selected	0	0.0%	6	5.4%	0.6525
	Not selected	8	100.0%	105	94.6%	
Issue warning	Selected	2	25.0%	46	41.4%	0.2060
	Not selected	6	75.0%	65	58.6%	
In-school suspension	Selected	2	25.0%	41	36.9%	0.2517
	Not selected	6	75.0%	70	63.1%	
Suspend from school	Selected	7	87.5%	96	86.5%	0.4042
	Not selected	1	12.5%	15	13.5%	
Refer to school administrator	Selected	8	100.0%	107	96.4%	0.7542
	Not selected	0	0.0%	4	3.6%	
Refer to counsellor	Selected	4	50.0%	65	58.6%	0.2539
	Not selected	4	50.0%	46	41.4%	
Encourage participation in an assistance, education, or cessation program	Selected	3	37.5%	70	63.1%	0.1087
	Not selected	5	62.5%	41	36.9%	
Confiscate substance	Selected	8	100.0%	105	94.6%	0.6525
	Not selected	0	0.0%	6	5.4%	
Place in detention	Selected	3	37.5%	30	27.0%	0.2423
	Not selected	5	62.5%	81	73.0%	
Alert police	Selected	5	62.5%	92	82.9%	0.1266
	Not selected	3	37.5%	19	17.1%	

***** Interpret results with caution given small cell counts. Fisher’s exact test (Freemon–Halton extension for R > 2) used to account for small cell sizes.

**Table 5 ijerph-17-05549-t005:** Student cannabis use by disciplinary options selected by secondary schools as potential approaches used in response to student violations of school substance use policies in Year 6 (2017/2018) of the COMPASS study.

		Current Cannabis Use	*p*-Value
*First-Offence Disciplinary Approach Selected*		%	
**Inform parents**	Selected	11.6%	0.4981
	Not selected	11.2%	
**Issue fine**	Selected	11.8%	0.1595
	Not selected	11.4%	
**Assign additional class work**	Selected	8.5%	<0.0001
	Not selected	11.8%	
**Require participation in an assistance, education, or cessation program**	Selected	10.7%	<0.0001
	Not selected	12.0%	
**Assign to help around school**	Selected	10.9%	0.2173
	Not selected	11.6%	
**Issue warning**	Selected	11.0%	0.0008
	Not selected	11.9%	
**In-school suspension**	Selected	9.8%	<0.0001
	Not selected	12.7%	
**Suspend from school**	Selected	11.6%	0.0350
	Not selected	10.8%	
**Refer to a school administrator**	Selected	11.6%	0.1955
	Not selected	10.6%	
**Refer to counsellor**	Selected	12.0%	<0.0001
	Not selected	10.7%	
**Encourage but not require participation an assistance, education, or cessation program**	Selected	12.0%	<0.0001
	Not selected	10.8%	
**Confiscate substance**	Selected	11.6%	0.0015
	Not selected	9.5%	
**Place in detention**	Selected	11.8%	0.1595
	Not selected	11.4%	
**Alert police**	Selected	11.3%	<0.0001
	Not selected	12.9%	
***Progressive discipline approach***	Always	11.4%	<0.0001
	Sometimes/Never	13.7%	

Note: Current cannabis use was defined as using cannabis at least once a month in the last 12 months.

**Table 6 ijerph-17-05549-t006:** Regression models for disciplinary approaches selected by schools as options used in response to school substance use policy violations and student cannabis use in Year 6 (2017/2018) of the COMPASS study (*n* = 60,384).

		Model 1 OR (95% CI)	Model 2 OR (95% CI)	Model 3 OR (95% CI)
Student-Level Characteristics				
Grade (ref: 9)	10	1.45 (1.30, 1.63) ***	1.45 (1.29, 1.62) ***	1.45 (1.29, 1.63) ***
	11	1.65 (1.44, 1.88) ***	1.65 (1.44, 1.89) ***	1.66 (1.45, 1.89) ***
	12	1.82 (1.58, 2.10) ***	1.83 (1.59, 2.12) ***	1.84 (1.60, 2.12) ***
	Other	0.83 (0.65,1.04)	0.83 (0.65,1.06)	0.84 (0.66, 1.07)
Sex (ref: Female)	Male	1.38 (1.28, 1.49) ***	1.39 (1.28, 1.50) ***	1.38 (1.28, 1.50) ***
Ethnicity (ref: White)	Non-white or mixed	1.10 (0.94, 1.28)	1.10 (0.96,1.28)	1.10 (0.96, 1.28)
Current Binge Drinking (ref: Non-binge drinker)		6.20 (5.67, 6.77) ***	6.24 (5.72, 6.82) ***	6.25 (5.72, 6.82) ***
Current Smoking (ref: Non-smoker)		11.09 (10.00, 12.30) ***	11.13 (10.09, 12.28) ***	11.11 (10.06, 12.26) ***
School-Level Characteristics				
*First Offence Disciplinary Approach Options:*				
Inform parents		-	0.92 (0.70, 1.21)	1.00 (0.75, 1.35)
Issue a fine		-	1.03 (0.74, 1.43)	1.04 (0.75, 1.44)
Assign additional class work		-	0.57 (0.38, 0.85) **	0.57 (0.38, 0.84) **
Require participation in a program^a^		-	0.98 (0.83, 1.16)	1.00 (0.84, 1.19)
Assign to help around school		-	0.91 (0.62, 1.32)	0.91 (0.64, 1.31)
Issue warning		-	1.01 (0.86, 1.17)	1.00 (0.86, 1.17)
In-school suspension		-	0.87 (0.74, 1.03)	0.88 (0.75, 1.05)
Suspend from school		-	1.20 (0.95, 1.51)	1.15 (0.94, 1.42)
Refer to school administrator		-	1.31 (1.04, 1.66) *	1.27 (1.00, 1.63)
Refer to counsellor		-	1.04 (0.89, 1.22)	1.00 (0.86, 1.17)
Encourage but do not require a program^a^		-	1.13 (0.98, 1.29)	1.16 (1.01, 1.33) *
Confiscate substance		-	1.21 (0.78, 1.86)	1.15 (0.75, 1.76)
Place in detention		-	1.16 (0.95, 1.43)	1.14 (0.93, 1.40)
Alert police		-	0.78 (0.64, 0.96) *	0.81 (0.67, 0.98) *
*Progressive Discipline Approach (Always)*		0.71 (0.56, 0.90) **	-	0.77 (0.62, 0.96) *

^a^ Assistance, education, or cessation program. Note: Current cannabis use was defined as using cannabis at least once a month in the last 12 months. * *p* < 0.05, ** *p* < 0.01, *** *p* < 0.001. Generalized linear random intercept models controlled for student (grade, sex, ethnicity, binge drinking and smoking) and school-level (province and school-area medium income) covariates and school clustering. Model 1 includes only school reported always use of the progressive discipline approach (vs. sometimes/never). Model 2 includes all first-offence disciplinary approach options without inclusion of school reported use of progressive discipline approach in the model. Model 3 includes all first-offence disciplinary approach options and school reported use of the progressive discipline approach in the model.

## Data Availability

COMPASS data are available for researchers upon request through successful completion and approval of the online COMPASS data usage application (https://uwaterloo.ca/compass-system/information-researchers).

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
