# Peer review of "Are School Substance Use Policy Violation Disciplinary Consequences Associated with Student Engagement in Cannabis?"

_ijerph, 2020, doi:10.3390/ijerph17155549_

Round 1
Reviewer 1 Report
This is a very interesting and extensive study based on data from the COMPASS study in Canada. All components of the study are very concise and clear and very well organised.
The manuscript addresses very interesting questions, and it clearly and fully presents the rationale, methodology, findings and conclusions drawn from the results derived. Relevant literature is also discussed either through original investigation references or through relevant review papers cited. The presentation of the results on the Tables is a major asset.
Overall, the findings are interesting and the manuscript is well prepared. There are only a few minor corrections/suggestions that would help further improve this manuscript:
- lines 56-59: please correct the font size.
- It is unclear why the researchers selected to include the "sometimes" answer to the progressive discipline approach in the "no" section. "Sometimes" is not the same as "never". The frequency and justification of this "sometimes" at school level could have also been investigated.
- line 202: "policies"
- line 215: do the authors mean "cessation"?
- lines 225-239: A clear explanation and justification is needed on why "alert police" was used as a reference for all analyses.
- line 273: Table 6.
- Table 6: Please fix the alignment of the Model Columns as they do not correspond exactly with the student-level characteristics.
- lines 387-389: It is unclear if this is related to more or less cannabis use. Please rewrite this sentence.
- line 393: "...less likely TO report...."
- line 469: Some funding information is missing in the [XXX] section on this sentence.
Author Response
Reviewer #1
This is a very interesting and extensive study based on data from the COMPASS study in Canada. All components of the study are very concise and clear and very well organised.
The manuscript addresses very interesting questions, and it clearly and fully presents the rationale, methodology, findings and conclusions drawn from the results derived. Relevant literature is also discussed either through original investigation references or through relevant review papers cited. The presentation of the results on the Tables is a major asset.
Overall, the findings are interesting and the manuscript is well prepared. There are only a few minor corrections/suggestions that would help further improve this manuscript:
Response: Thank you for your feedback and suggestions to help us improve our manuscript. Please find the corrections addressed appropriately in the document and outlined below.
1. lines 56-59: please correct the font size.
Response: Thank you. This correction has been fixed in the document.
2. It is unclear why the researchers selected to include the "sometimes" answer to the progressive discipline approach in the "no" section. "Sometimes" is not the same as "never". The frequency and justification of this "sometimes" at school level could have also been investigated.
Response: Thank you for this feedback. “Sometimes” and “Never” were collapsed together because of the nature of the data that was collected. Only one school selected “never” using the progressive discipline approach. Whereas, 7 schools indicated using the progressive discipline approach “sometimes”. Using the progressive discipline approach “sometimes” could also mean schools have only used this approach once. “Always” using this approach would mean the school uses it consistently, which would be different from using the approach once and a while. Also, unfortunately, with only one school indicating “never”, we did not have adequate power to examine differences between never, sometimes, and always as three separate categories. Within the document, on lines 154-155, an explanation has been provided stating “Since only one school indicated “never” using a progressive disciplinary approach, responses were collapsed into “no” if the school reported “sometimes” or “never”.
Unfortunately, we do not have data for the frequency at which schools actually use the progressive discipline approach. This would be something to consider investigating further in future research.
3. line 202: "policies"
Response: Thank you for this correction. The document has been updated to fix this error.
4. line 215: do the authors mean "cessation"?
Response: The notes under the table have been updated to read “cessation”.
5. lines 225-239: A clear explanation and justification is needed on why "alert police" was used as a reference for all analyses.
Response: We were looking to investigate patterns in how the schools completed the disciplinary response measures. Of interest was whether schools that selected alerting the police as a first-offence disciplinary approach also selected more punitive approaches overall. To provide further rationale, we have added on lines 180-181: “To investigate patterns in how the schools completed the disciplinary response measures, and given that alerting the policy is a more severe disciplinary response, and then we then compared responses based on their selection of “alert police” among their choice of first-offence approaches.” and to lines 401-405: “Alerting the police is not a requirement of schools when they catch students violating school substance use policies in Canada. It is often up to the discretion of the principal to decide whether to involve the police. Given that alerting the police is a more severe disciplinary response, we explored the possibility that schools that selected this option may be different from those that did not.”
6. line 273: Table 6.
Response: Thank you for correcting this error. It has been updated in the document on line 273.
7. Table 6: Please fix the alignment of the Model Columns as they do not correspond exactly with the student-level characteristics.
Response: All columns in this table have been fixed to better align with the student-level characteristics.
8. lines 387-389: It is unclear if this is related to more or less cannabis use. Please rewrite this sentence.
Response: Lines 387-389 have been rewritten to now read: “While cross-sectional, these results compliment previous research indicating that the use of less punitive and more supportive approaches are associated with less student cannabis use (14) (9) (3).”
9. line 393: "...less likely TO report...."
Response: Thank you. This has been updated in the document on line 393.
10. line 469: Some funding information is missing in the [XXX] section on this sentence.
Response: Thank you for noticing this error. We have updated the funding information in the revised manuscript.
Reviewer 2 Report
Very nice article. I am wondering how the results of this study would be now that Cannabis is legal for adults in Canada.
The youth are the most vulnerable group of users and to control the use and limit it is a very difficult task. We are moving away from punitive responses but yet, any form of substance use disorder will require some structure and expectations. In particular in the adolescent structure and limit setting will be helpful for the rest of their lifes.
Your graphs in the result section are not easy to read and understand but appears statistically correct. Consider using graphs for improved optic appearance and easier read for the future.
Author Response
Reviewer #2:
Very nice article. I am wondering how the results of this study would be now that Cannabis is legal for adults in Canada.
The youth are the most vulnerable group of users and to control the use and limit it is a very difficult task. We are moving away from punitive responses but yet, any form of substance use disorder will require some structure and expectations. In particular in the adolescent structure and limit setting will be helpful for the rest of their lifes.
Your graphs in the result section are not easy to read and understand but appears statistically correct. Consider using graphs for improved optic appearance and easier read for the future.
Response: Thank you for your feedback and suggestions. In the future, we will consider using graphs instead, to better articulate the findings to readers. We have made changes to the tables in the manuscript to make them clearer. We are pleased the reviewer sees value in the manuscript.
Reviewer 3 Report
Well done manuscripts. Methods appropriately described. No clear statistical violations. Authors present evidence-based conclusions without generalizing their findings in excess.
Author Response
Reviewer #3:
Well done manuscripts. Methods appropriately described. No clear statistical violations. Authors present evidence-based conclusions without generalizing their findings in excess.
Response: Thank you for your review of our manuscript and feedback.